# ED Formula, a Complex of *Ecklonia cava* and *Chrysanthemum indicum*, Ameliorates Airway Inflammation in Lipopolysaccharide-Stimulated RAW Macrophages and Ovalbumin-Induced Asthma Mouse Model

**DOI:** 10.3390/ph16081185

**Published:** 2023-08-21

**Authors:** Hyun Kang, Chan-Hwi Park, Sang-Oh Kwon, Sung-Gyu Lee

**Affiliations:** 1Department of Medical Laboratory Science, College of Health Science, Dankook University, Cheonan-si 31116, Chungnam, Republic of Korea; hkang@dankook.ac.kr (H.K.); cksgnl1014@naver.com (C.-H.P.); 2S&D Co., Ltd., 473, Mansu-ri, Osong-eup, Heungdeok-gu, Cheongju-si 28156, Chungcheongbuk-do, Republic of Korea; so-kwon0004@hanmail.net

**Keywords:** *Ecklonia cava*, *Chrysanthemum indicum*, airway inflammation, asthma, dieckol, mouse models

## Abstract

*Ecklonia cava* (*E. cava*) and *Chrysanthemum indicum* Linne (*C. indicum*) are natural raw materials known to have beneficial effects on inflammatory-related diseases, as evidenced by various sources in the literature. This study aimed to investigate the airway-protective effects of a formulation called ED, comprising *E. cava* and *C. indicum*, by evaluating its potential anti-inflammatory properties. Methods: The major components of ED were analyzed using high-performance liquid chromatography (HPLC) and its anti-inflammatory activity was assessed in RAW 264.7 cells through measurements of nitric oxide’s (NO) inhibitory effect, cyclooxygenase (COX)-2 protein expression, and the mitogen-activated protein kinase (MAPK) signaling pathway. Additionally, the anti-inflammatory effect of ED was evaluated in an ovalbumin-induced asthma model by measuring cytokine levels in serum, bronchoalveolar lavage fluid (BALF), and lung tissue. Through HPLC analysis, the major components of ED, dieckol and luteolin, were identified. ED demonstrated no cytotoxicity and effectively reduced NO production in lipopolysaccharide (LPS)-induced RAW 264.7 cells. Moreover, ED downregulated COX-2 expression through the MAPK signaling pathway in LPS-induced RAW 264.7 cells. In the ovalbumin-induced asthma model, the ED-treated group exhibited reduced levels of inflammatory cytokines in lung tissue. Furthermore, the ED-treated group showed a decrease in the number of inflammatory cells in BALF and lower serum interleukin (IL)-6 levels compared to the ovalbumin-treated group. These results suggest that ED has the potential to be a novel therapeutic agent for improving inflammatory respiratory diseases.

## 1. Introduction

Asthma is a common chronic inflammatory disease characterized by the presence of inflammatory cells in the airways and affects approximately 300 million people worldwide. It can be triggered by allergens such as pollen, house dust, animal dander, inhalers, food, and air pollutants [1]. Typical symptoms of asthma include shortness of breath, difficulty breathing, and coughing, which are associated with airway inflammation, influx of inflammatory cells, excessive mucus production, and airway hyper-responsiveness [1]. Asthma is a global health problem with increasing prevalence and significant health-related costs [2,3]. Understanding the mechanisms underlying asthma development and developing prevention or improvement strategies is crucial.

Most early-stage asthma patients are treated with steroid inhalation, long-acting beta 2 agonists (LABA), and leukotriene modifiers, while severe asthma patients often require repeated use of systemic steroids [4,5]. Despite ongoing research on asthma medications and treatment approaches, many patients continue to suffer from the burdens of asthma, leading to interest in the development of natural drugs due to dissatisfaction with treatment outcomes or the occurrence of side effects [6,7,8].

*Ecklonia cava* (*E. cava*), a seaweed belonging to the brown alga (Laminariaceae) and widely distributed in coastal areas of Korea, China, and Japan, contains a variety of bioactive compounds, including phlorotannins and fucoidans [9]. Phlorotannins belong to the group of polyphenolic compounds that are particularly abundant in seaweeds. These compounds provide the brown coloration of *E. cava* and are produced by seaweeds as a means of self-defense and survival. Eckol, dieckol, 6,6’-bieckol, and phlorofucofuroeckol have been identified as major phlorotannin components in *E. cava* [10]. Phlorotannins exhibit potent antioxidant activity [11]. *Chrysanthemum indicum* Linne (*C. indicum*) is a perennial herbaceous plant native to Korea and belongs to the *Chrysanthemum* genus in the Compositae family. It typically blooms with yellow flowers from September to October [12]. *C. indicum* contains a significant amount of flavonoid compounds as its major constituents, with apigenin, luteolin, acacetin, and other flavonoid derivatives being predominantly present. Among them, luteolin has been reported to possess anticancer and anti-inflammatory activities [13,14,15]. Our laboratory has been establishing a library of natural products for a long time, so we conducted screening for airway inflammation alleviation for several years, found seaweed, *E. cava*, Asteraceae plants, and *C. indicum* to be effective, and conducted more specific experiments. As a result of screening for ways to ameliorate airway inflammation, we found that the most important biomarker was inhibiting glycoprotein MUC5AC in human airway epithelial cells [16,17,18].

We investigated the effects of a formula composed of *E. cava* and *C. indicum* on airway inflammation in an ovalbumin (OVA)-induced asthma mouse model. According to the high-performance liquid chromatography (HPLC) analysis that was conducted, the active compounds of *E. cava* and *C. indicum*, dieckol and luteolin, were presented as the ED compounds. ED, along with its two constituent compounds, reduced the levels of nitric oxide (NO) and cyclooxygenase (COX)-2 in lipopolysaccharide (LPS)-induced RAW 264.7 cells, indicating an anti-inflammatory effect. Particularly, ED strongly suppressed inflammation through the NF-κB and MAPK pathways. In vivo results demonstrated that ED reduced the proliferation of inflammatory cells in bronchoalveolar lavage fluid (BALF) and lung tissue, and it decreased the expression of pro-inflammatory cytokine genes. These results suggest that ED could be considered as an excellent therapeutic approach for the prevention of asthma.

## 2. Results

### 2.1. Dieckol and Luteolin Content in ED

The analysis of the content of the active compounds, dieckol and luteolin, in the ED is shown in Figure 1. On the HPLC chromatogram, dieckol exhibited a retention time of approximately 25.49 min with a content of 9.5 mg/g. Luteolin, on the other hand, had a retention time of approximately 15.11 min and a content of 1.5 mg/g.

### 2.2. ED and Its Active Compounds Suppress LPS-Induced Inflammation in RAW 264.7 Cells

Prior to investigating the anti-inflammation effects of ED and its active compounds, a 3-(4,5-dimethylthiazol-2-yl)-2,5-diphenyltetrazolium bromide (MTT) assay was performed to investigate the dose-dependent cell viability effect in RAW 264.7 cells. Cells were treated with different concentrations of ED (25, 50, 100, 200, and 400 μg/mL), dieckol (5, 10, and 20 μg/mL), and luteolin (5, 10, and 20 μg/mL) for 24 h, and then an MTT analysis was performed. According to the results, ED and its main active compounds, which were tested separately, had no cytotoxic effects at any of the concentrations tested (Figure 2A).

We investigated whether ED and its active compounds have the ability to regulate the inflammatory response of macrophages in response to LPS stimulation. We utilized a Griess reagent to determine the level of NO in the supernatants collected from cells treated with various concentrations of ED and its active compounds and then stimulated with LPS for 24 h. The results showed that treatment with ED and its active compounds potently inhibited LPS-induced production and secretion of NO in RAW 264.7 cells (Figure 2B).

We next examined the expression of COX-2, a major pro-inflammatory mediator. As shown in Figure 2C–E, the protein expression levels of COX-2 were significantly inhibited by ED and its active compounds dose-dependently from activated RAW 264.7 cells.

### 2.3. ED Inhibits LPS-Induced Nuclear Factor-kappa B (NF-κB) and Mitogen-Activated Protein Kinase (MAPK) Signaling Activation

The effect of ED on the expression of NF-κB in relation to anti-inflammatory activity was investigated. It was observed that the increased expression of NF-κB induced by LPS was significantly reduced in a concentration-dependent manner by treatment with ED (Figure 3A). Measurement of phosphorylated NF-κB p65 in the cytoplasm revealed an increase upon LPS stimulation, which decreased with increasing concentrations of ED treatment (Figure 3B). Therefore, it is suggested that the translocation of NF-κB p65 into the nucleus, induced by LPS, was inhibited by ED. The influence of ED on the expression regulation of MAPKs in relation to anti-inflammatory activity was also investigated. The results showed that the increased expression of phosphorylated JNK, ERK, and p38 induced by LPS stimulation was significantly inhibited in a concentration-dependent manner by ED treatment (Figure 3C–E). Thus, it was observed that ED exhibited anti-inflammatory activity in LPS-induced RAW 264.7 cells by suppressing the expression of the inflammatory mediator COX-2 and the generation of NO through the inhibition of NF-κB and MAPKs’ activities.

### 2.4. ED Ameliorates Airway Inflammation in OVA-Induced Asthmatic Mice

#### 2.4.1. Treatment with ED Reduced the Number of Total Cells in BALF

For each group, BALF cytospin slides were stained using a Diff-Quick kit, and the cells were observed under a microscope (Figure 4). The total number of cells in the OVA group significantly increased compared to that in the normal group (# *p* < 0.05). In contrast, the ED300 and Bronpass300 groups significantly decreased the number of total cells in BALF compared to the OVA group (** *p* < 0.01).

#### 2.4.2. Effects of ED on Lung Histological Changes

Inflammatory cell infiltration and mucus secretion are major characteristics of asthma. Histological changes in lung tissue were detected using hematoxylin and eosin (H&E) as well as periodic acid-Schiff (PAS) staining analysis. As shown in Figure 5, lung tissues in the OVA group exhibited significant inflammatory cell infiltration. Additionally, PAS staining indicated that mice in the OVA group had excessive mucus production. However, the ED300 group showed considerable improvement in OVA-induced inflammatory cell infiltration and significantly suppressed excessive mucus secretion, which was similar to the effects observed in the Bronpass300 group.

#### 2.4.3. Effects of ED on Inflammatory Cytokine Interleukin (IL)-6 in Serum

In order to investigate the effect of ED on the inflammatory cytokine IL-6 in OVA-induced asthma mice, the IL-6 levels in the serum were analyzed. The IL-6 levels in the OVA group significantly increased at 77.03 ± 18.05 pg/mL compared to the normal group (*## p* < 0.01). However, in the ED300 and Bronpass300 groups, the IL-6 levels decreased to 36.87 ± 1.07 and 53.39 ± 24.00 pg/mL, respectively. Particularly, the ED300 group exhibited a superior decrease in IL-6 levels compared to the positive control group, Bronpass300, at the same administered dose (300 mg/kg/day) (Figure 6).

#### 2.4.4. Effects of ED on the mRNA Expression Levels of Inflammatory Mediators in the Lung Tissue

In this study, we examined the mRNA expression levels of iNOS, tumor necrosis factor (TNF)-α, IL-1β, and COX-2 in lung tissues obtained from the normal group, OVA group, ED150 group, ED300 group, and Bronpass300 group. We compared these expression levels to those of the OVA group. Our results demonstrated that the ED150, ED300, and Bronpass300 groups exhibited significantly decreased mRNA expression levels of iNOS, TNF-α, IL-1β, and COX-2 when compared to the OVA group (Figure 7).

## 3. Discussion

Asthma is a chronic respiratory disease characterized by airway inflammation and hypersensitivity [19]. Steroids are the most commonly prescribed medication for asthma, but they come with side effects such as hypertension, hyperlipidemia, peptic ulcers, myopathy, growth inhibition in children [20], and immunosuppressive effects [21]. As a result, recent research has focused on identifying pharmacologically active compounds in natural resources. The approach of using natural drugs for the treatment of chronic asthma offers the potential to reduce unwanted side effects associated with long-term drug administration.

This study provides evidence that ED, a formula of *E. cava* and *C. indicum*, exerts an inhibitory effect on airway inflammation in an OVA-induced asthma model.

Dieckol, a major component of *E. cava*, belonging to the phlorotannin class, has been reported to possess anti-inflammatory, antioxidant, antidiabetic, antimicrobial, and anticancer properties [9,22,23,24]. *C. indicum* is a traditional medicinal herb used in Korea and China for treating various inflammatory-related diseases and is known for its high efficacy and low toxicity [25]. Recent pharmacological research has shown that the extract of *C. indicum* inhibits the NF-κB dependent pathway, reducing inflammatory mediators in LPS-induced RAW 264.7 cells [15], and blocking the synthesis of inflammatory cytokines such as NO and prostaglandin E_2_ (PGE_2_) [26]. This plant contains mainly polyphenolic and flavonoid family compounds, among which luteolin has been suggested to play a role in treating inflammation-related diseases by reducing reactive oxygen species (ROS) and pro-inflammatory cytokines [27,28].

Macrophages play an important role in inflammation and are the primary targets of LPS action. LPS-induced RAW 264.7 cells have been widely used as a model to investigate inflammatory responses [29]. Activation of LPS-induced RAW 264.7 cells significantly upregulates pro-inflammatory mediators such as NO and ROS [30]. NO plays a dominant role in all stages of inflammation and can be an important target in the treatment of chronic inflammation [31]. In this study, we found that ED and its active components, dieckol and luteolin, showed significant inhibitory activity against NO production in LPS-induced RAW 264.7 cells. COX-2 is closely related to inflammation because it produces PGE_2_ [32]. Therefore, the effects of ED and its active components on COX-2 expression in LPS-induced RAW 264.7 cells were investigated. We found that ED and its active constituents significantly inhibited the LPS-induced production of NO. Several studies have reported that NO production and COX-2 are regulated by similar signaling pathways, such as activator protein (AP)-1 and NF-κB [33,34]. Our results also confirmed that ED and its active components suppressed COX-2 expression. Therefore, it was confirmed that ED and its active components, dieckol and luteolin, simultaneously inhibit NO production and regulate COX-2 expression in LPS-induced RAW 264.7 cells. Identifying the compounds responsible for a specific biological effect in natural product extracts can be a significant challenge as these extracts may contain a number of unknown constituents in varying abundances [35]. Several studies have demonstrated that the overall activity of botanical extracts often arises from mixtures of compounds with synergistic, additive, or antagonistic effects [36]. In this study, we conducted an experiment comparing the anti-inflammatory effect of ED with dieckol and luteolin purified from each raw material. As a result of the experiment, ED showed superior efficacy in inhibiting COX-2 expression compared to purified dieckol and luteolin. We suggest that the anti-inflammatory activity of these EDs may target multiple molecular targets simultaneously. From these results, the identification of the mechanism of action and synergistic effect of ED and unknown components of ED suggests that more interesting research can be conducted in the near future.

NF-κB and MAPKs, including JNK, ERK, and p38, play important roles in cytokine production and cell differentiation and growth [37]. LPS-induced RAW 264.7 cells induce phosphorylation of MAPK and are involved in the activation of transcription factors including AP-1, cAMP response element-binding protein (CREB), signal transducer and activator of transcription1 (STAT1), and NF-κB; they produce inflammatory cytokines [37,38,39]. Previous reports have shown that effective drugs against inflammation are regulated by the downregulation of MAPK and NF-κB in LPS-induced RAW264.7 cells [40]. As is consistent with previous reports, the present study showed that ED reduced the phosphorylation of ERK and p38 and reduced the activation of NF-κB in LPS-induced RAW 264.7 cells. Our results suggest that ED is an anti-inflammatory natural compound that inhibits NO production and COX-2 expression by downregulating the LPS-induced MAPK and NF-κB signaling pathways.

We conducted further in vivo studies to validate the anti-inflammatory effects of ED and to confirm the physiological relevance of the in vitro findings. For this purpose, we used a mouse model of OVA-induced asthma. Asthma is characterized by the presence of cellular infiltrates, mainly composed of monocytes and eosinophils. Eosinophils and leukocytes are crucial cell types involved in airway inflammation, found not only in the airway walls but also in the BALF [41]. To assess the impact of ED, we collected BALF from the lungs of BALB/c mice that were exposed to OVA. This allowed us to evaluate the total number of leukocytes and eosinophils. We observed a significant increase in the number of leukocytes and eosinophils in the BALF of the OVA group compared to the normal group. However, treatment with ED at a dose of 300 mg/kg/day led to a significant reduction in leukocyte and eosinophil counts in BALF when compared to the OVA group. These effects were comparable to those observed in the positive control group treated with Bronpass300, a known asthma agent made using a medicinal herb. Photomicrographs of the BALF and histological analysis of lung tissue demonstrated that ED effectively suppressed eosinophil infiltration and mucus secretion in the lungs (Figure 4 and Figure 5). Based on these findings, we can infer that ED exhibits therapeutic potential by alleviating airway obstruction and eosinophilic inflammation associated with allergic asthma. It achieves this by inhibiting the infiltration of inflammatory cells into the lungs and reducing excessive mucus production. Several previous studies by Tiwari et al. [42] and Kim et al. [43] have reported the protective effects of various plant extracts and plant-derived compounds on airway inflammation. Additionally, Lee et al. [44] demonstrated that the 80% ethanolic extract of *Petasites japanicus* (50, 250, and 500 mg/kg) significantly and dose-dependently inhibited eosinophil infiltration and mucus overproduction induced in the airways. Moreover, it was shown to reduce reactive oxygen species (ROS) production and the release of inflammatory cytokines into lung tissues. These findings provide supporting evidence for our current study.

The levels of the inflammatory cytokine IL-6 were assessed in the serum of a mouse model of OVA-induced asthma. In comparison to the normal group, the OVA group exhibited significantly elevated levels of IL-6. However, treatment with ED at a dose of 300 mg/kg/day (ED300 group) resulted in a significant reduction in IL-6 levels. IL-6 is a pleiotropic protein produced by various cells in the innate and adaptive immune systems, as well as structural cells such as epithelial cells. It functions as a cytokine. There is mounting evidence implicating IL-6 signaling in the pathogenesis of asthma. Increased IL-6 levels have been observed in the serum [45], induced sputum [46], and BALF [47] of asthma patients and are associated with compromised lung function and asthma severity [48]. Consequently, these findings suggest that ED effectively attenuates pulmonary inflammation by inhibiting IL-6 production.

Inflammatory cell infiltration into lung tissue is a characteristic feature of OVA-induced asthma. The excessive presence of inflammatory cells promotes the production of inflammation mediators, including cytokines and chemokines, thereby intensifying the inflammatory response within the lungs [49]. Through our study, we verified that treatment with ED significantly reduced the number of inflammatory cells in the OVA-induced asthma mouse model. Consequently, we examined the mRNA expression levels of key inflammatory factors, namely, iNOS, TNF-α, IL-1β, and COX-2, in lung tissue. As was consistent with the histological observations, the OVA group exhibited increased mRNA expression of iNOS, TNF-α, IL-1β, and COX-2. However, in the group treated with ED, we observed a noticeable decrease in the mRNA expression levels of these inflammatory factors.

## 4. Materials and Methods

### 4.1. Preparation of ED

ED is a mixture of *E. cava* ethanol extract and *C. indicum* concentrate, which was provided by S&D Co., Ltd. (Cheongju-si, Chungcheongbuk-do, Korea). *E. cava* ethanol extract is a health-functional ingredient that is individually recognized by the Korea food and drug administration approval (S&D Co., Ltd., Recognition No. 2015-6). *E. cava* ethanol extract was purified from the ethanol extract of the brown alga *E. cava*, then manufactured to 60 mg/g dieckol. *C. indicum* was extracted, filtered, and concentrated, and then, after adding and mixing *E. cava* extract and an excipient (dextrin), ED (Lot. SD-ED-001) was prepared using a spray dryer (inlet temp: 185,210 °C, outlet temp: 8598 °C) and used in the experiment.

### 4.2. Quantification of Active Compounds of ED

#### 4.2.1. Standards, Reagents, and Materials

Dieckol purified from S&D Co., Ltd. and luteolin purchased from Sigma-Aldrich Co. (St. Louis, MO, USA) were used as standard compounds. Methanol, acetic acid, and trifluoroacetic acid (HPLC grade) were bought from Merck (Darmstadt, Germany). Deionized water (DW) was prepared using a Millipore Milli-Q purification system (Millipore, Bedford, MA, USA).

#### 4.2.2. HPLC Analysis

The analysis was performed using an Agilent 1200 series instrument (Agilent Technologies, Inc., Santa Clara, CA, USA) with a Supelco Discovery C18 column (4.6 mm × 250 mm, 5 μm) as the analytical column. For dieckol analysis, approximately 50 mg of ED was dissolved in 10 mL of dimethyl sulfoxide (DMSO). Subsequently, 40 mL of methanol was added to the solution and ultrasonic extraction was performed. After cooling the solution to room temperature (RT), it was filtered through a filter and used for the experiment. Regarding luteolin analysis, around 800 mg of ED was dissolved in 20 mL of methanol, followed by ultrasonic extraction. The resulting solution was then filtered through a filter before being utilized in the experiment. The analysis conditions were as described in Table 1.

### 4.3. In Vitro Anti-Inflammatory Activity of ED in LPS-Induced RAW 264.7 Macrophages

#### 4.3.1. Cell Culture

RAW 264.7 murine macrophage cells were purchased from the Korean Cell Line Bank (KCLB; Seoul, Korea). The cells were incubated at 37 °C in 5% CO_2_ in Dulbecco’s modified Eagle’s media (DMEM; Gibco-BRL, Carlsbad, CA, USA), supplemented with 100 U/mL penicillin, 100 µg/mL streptomycin, and 10% fetal bovine serum (FBS; Gibco-BRL).

#### 4.3.2. Cell Viability Assay

To determine cell viability, an MTT analysis was performed. RAW 264.7 cells were cultured in a 96-well plate at a density of 1 × 10^5^ cells/well and incubated for 24 h. Subsequently, the cultured cells were treated with ED (25, 50, 100, 200, and 400 μg/mL), dieckol (5, 10, and 20 μg/mL), or luteolin (5, 10, and 20 μg/mL) for 24 h. Afterward, the MTT solution was added to each well for a final concentration of 0.5 mg/mL, and the cells were incubated at 37 °C in a CO_2_ incubator for 4 h. After removing the supernatant, the formed formazan was dissolved in 100 μL of dimethyl sulfoxide (DMSO) and mixed for 10 min. The optical density of the DMSO solution was measured at a wavelength of 550 nm using a microplate reader (Bio-Rad, Hercules, CA, USA). The absorbance of the formazan formed by the cells in the control group (untreated group) demonstrated 100% cell viability, and the cell viability of the other groups was expressed as a percentage of the surviving cells obtained compared to the control group.

#### 4.3.3. Measurement of NO Production

We measured the concentration of NO in the culture medium using the Griess reaction. RAW 264.7 cells were cultured in a 96-well plate at a density of 1 × 10^5^ cells/well and incubated for 24 h. The cells were then treated with ED (25, 50, 100, 200, and 400 μg/mL), dieckol (5, 10, and 20 μg/mL), or luteolin (5, 10, and 20 μg/mL) at different concentrations in the presence of LPS (Sigma-Aldrich) at a concentration of 100 ng/mL for 24 h. Each culture medium (100 μL) was mixed with an equal volume of Griess reagent (Sigma-Aldrich) at RT for 10 min. The absorbance was measured at a wavelength of 540 nm using a microplate reader (Bio-Rad). The concentration of NO in the culture medium was determined based on a standard curve of sodium nitrite (Sigma-Aldrich).

#### 4.3.4. Western Blot Analysis

For Western blot analysis, RAW 264.7 cells were seeded into 6-well plates at a density of 5 × 10^5^ cells/well and cultured for 24 h. Then, the cells were treated with samples prepared at different concentrations in a serum-free medium for 24 h. The cells were then lysed in RIPA lysis buffer (Sigma-Aldrich Co.) on ice. Protein concentrations were determined using the BCA Protein Assay Kit (Thermo Scientific, Rockford, IL, USA). Equal amounts of protein (20 μg) were separated by SDS-PAGE on 10% gels and subsequently transferred onto PVDF membranes for 2 h using the Mini Gel Tank system (Invitrogen, Gaisburg, MD, USA). The PVDF membranes were blocked with 5% skim milk in TBST for 1 h, followed by washing with TBST. Primary rabbit monoclonal antibodies (Cell Signaling Technology, Danvers, MA, USA) against COX-2, P65, p-p65, JNK, p-JNK, ERK, p-ERK, p38, p-p38, β-Actin, and Lamin B1 were then added at a 1:1000 dilution and incubated overnight at 4 °C. The membranes were washed three times for 10 min with TBST and then incubated with an anti-rabbit HRP-linked antibody (Cell Signaling Technology, Danvers, MA, USA) diluted at 1:2000 for 2 h at RT. Protein bands were visualized using the ECL substrate reagent (Bio-Rad) with the chemiluminescence imaging system (WSE-6100 Luminograph, ATTO, Tokyo, Japan). Densitometry analysis was performed using Image J software (Version 1.53t, U.S. National Institutes of Health, Bethesda, MD, USA) with β-actin and Lamin B1 as loading controls.

### 4.4. Evaluation of Anti-Airway Inflammation Activity of ED in OVA-induced Asthma Mouse Model

#### 4.4.1. Experimental Animals

Male BALB/c mice aged 6–8 weeks (weighing 25–30 g) were purchased from DBL Co., Ltd. (Eumsung-gun, Chungcheongbuk-do, Korea). The mice were maintained under standard laboratory conditions with a humidity of 50–60% and a temperature of 23 ± 1 °C. They were provided with a standard pellet diet and water ad libitum. All experimental procedures were approved by the Dankook University Institutional Animal Care and Use Committee (DKU IACUC Approval No: DKU-22-031).

#### 4.4.2. Animal Sensitization and Treatment

The animals were randomly divided into five groups (5 mice/group). Asthmatic mice were sensitized with 100 μL of chicken egg ovalbumin (OVA; Grade V; 20, 40 μg/mL) absorbed with aluminum hydroxide (Sigma-Aldrich Co., 2 mg/mL) intraperitoneally (i.p.) on days 1 and 14, and then challenged with aerosolized 1% OVA on days 21–26 for 60 min each day. The normal group received saline only, the OVA group was OVA sensitive and challenged, and the ED150 and ED300 groups received different concentrations of ED (ED 150 and 300 mg/kg). The Bronpass300 group served as a positive control group, receiving 300 mg/kg of Bronpass Table (p.o., Hanlim Pharm. Co., Ltd., Yongin-si, Korea). ED and Bronpass Tables were made into a suspension in DW and administered orally, starting from day 15, once daily for 12 days. On the 28th day, all animals were anesthetized with isoflurane inhalation, and blood, BALF, and lung tissue were separated and used for the experiment (Figure 8).

#### 4.4.3. Measurement of Inflammatory Cell Counts in BALF

After 48 h had passed since the last OVA challenge, the mice were anesthetized using isoflurane and a tracheostomy was performed. To obtain BALF, cold PBS (0.8 mL) was injected into the lungs and collected through a cannula. The collected BALF was subjected to centrifugation. The precipitated cells were evaluated by counting the cell numbers using a minimum of five fields on a hemocytometer after excluding dead cells with Trypan Blue staining. The cell concentration within 200 μL of BALF was adjusted to 2 × 10^4^, and centrifugation was performed using a cytospin centrifuge (Thermo Fisher Scientific Inc., Waltham, MA, USA) at 200 g for 10 min. The slides were air-dried and stained using Diff-Quick staining reagent (Thermo Fisher Scientific Inc.) following the manufacturer’s instructions.

#### 4.4.4. Determination of IL-6 in Serum

Whole blood was collected through the intraorbital vein at the euthanasia time point, as well as the serum was collected by centrifugation using BD Microtainer Tubes (Becton Dickinson, Franklin Lakes, NJ, USA). The serum IL-6 levels were quantified using the ELISA kit (Mouse IL-6 Immunoassay, R&D Systems, Minneapolis, MN, USA), following the manufacturer’s instructions.

#### 4.4.5. Histopathological Evaluation of Lungs

Lungs were isolated from the mice during sacrifice and fixed in a 10% formalin solution. After fixation, paraffin sections 5 μm thick were cut, and inflammatory cell infiltration and pulmonary fibrosis were measured by H&E staining. Additionally, lung tissues were stained with PAS staining to examine goblet cell proliferation and mucus overproduction [50].

#### 4.4.6. Reverse Transcription-Polymerase Chain Reaction (RT-PCR)

After homogenizing the lung tissue, total RNA was extracted using TRIzol reagent (Thermo Fisher Scientific) and then converted to cDNA using the ImProm-II Reverse Transcription System kit (Promega Corporation, Fitchburg, WI, USA). PCR amplification was performed using the PCR PreMix kit (Bioneer, Daejeon, Korea) with the addition of sense primers (20 pmole/mL) and antisense primers (20 pmole/mL). The primer sequences are shown in Table 2. The PCR products were electrophoresed on a 1.2% agarose gel.

### 4.5. Statistical Analysis

The experimental results were expressed as the mean ± standard deviation (S.D.), and statistical analysis was conducted using analysis of variance (ANOVA). Duncan’s multiple range test was performed to determine significant differences between groups. In this study, a *p* < 0.05 was considered statistically significant.

## 5. Conclusions

In this study, we confirmed that ED improves respiratory inflammation both in vitro and in vivo. ED is a formulation of *E. cava* and *C. indicum*, and it was found to dose-dependently decrease NO production as a non-cytotoxic drug. Additionally, it reduced COX-2 expression in LPS-induced RAW 264.7 cells. ED reduced the phosphorylation of ERK and p38, as well as the activation of NF-κB in the LPS-induced RAW 264.7 cells. Our results suggest that ED is an anti-inflammatory natural compound that inhibits NO production and COX-2 expression by downregulating LPS-induced MAPK and NF-κB signaling pathways. ED also decreased inflammatory cells in the BALF of mice in an OVA-induced asthma model and reduced serum levels of IL-6. Furthermore, our histological analysis of mouse lung tissue demonstrated reduced infiltration of inflammatory cells and mucus formation in the ED-treated group. These findings indicate that ED has the potential as a novel drug to improve respiratory inflammation.

## Figures and Tables

**Figure 1 pharmaceuticals-16-01185-f001:**
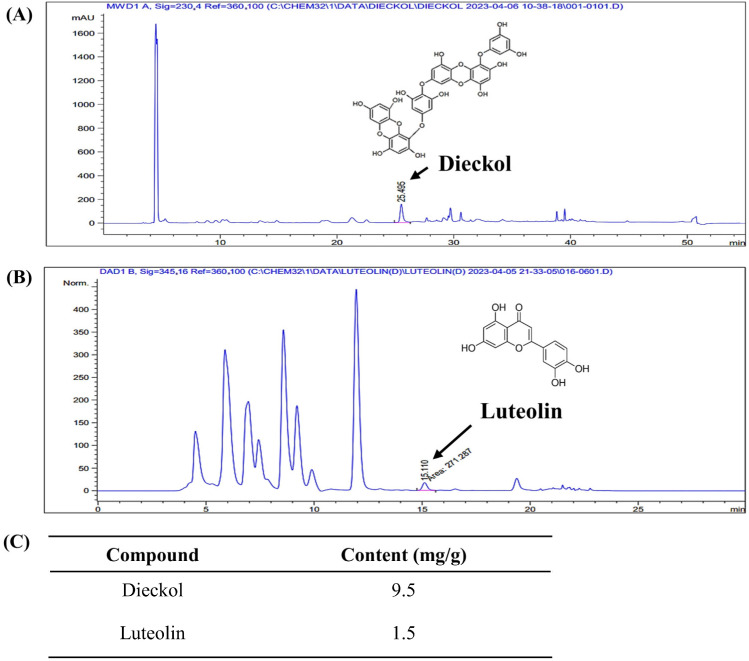
(**A**,**B**) HPLC spectrum of ED. (**C**) Dieckol and luteolin content in ED (unit: mg/g ED).

**Figure 2 pharmaceuticals-16-01185-f002:**
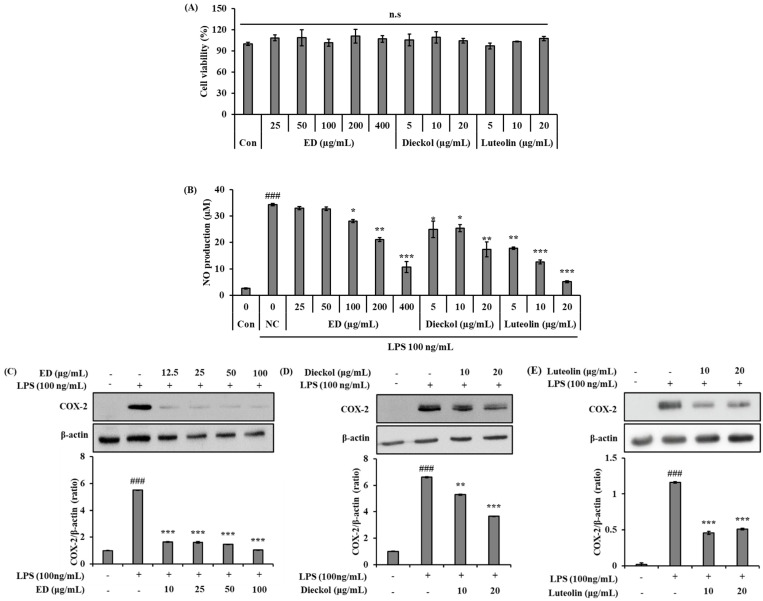
Effects of ED and its active compounds on the viability and inflammation of RAW 264.7 cells. (**A**) The bars the cell viability, in percent, of cells treated with ED and its active compounds with the various concentrations for 24 h, then subjected to MTT assay. (**B**) Cells were incubated with LPS (100 ng/mL) with or without ED and its active compounds for 24 h. Secretions of NO were determined using a Griess reagent. (**C**–**E**) Western blot analysis for the inhibitory effects of ED and its active compounds on the protein expression level of COX-2. β-actin was detected and used as an internal control. The average value of three independent experiments is shown. All data are expressed as the mean ± SD of the experiment. ### *p* < 0.001 compared to the control group; * *p* < 0.05, ** *p* < 0.01, and *** *p* < 0.001 compared to the LPS control group. n.s.: not statistically significant. See also Appendix A.

**Figure 3 pharmaceuticals-16-01185-f003:**
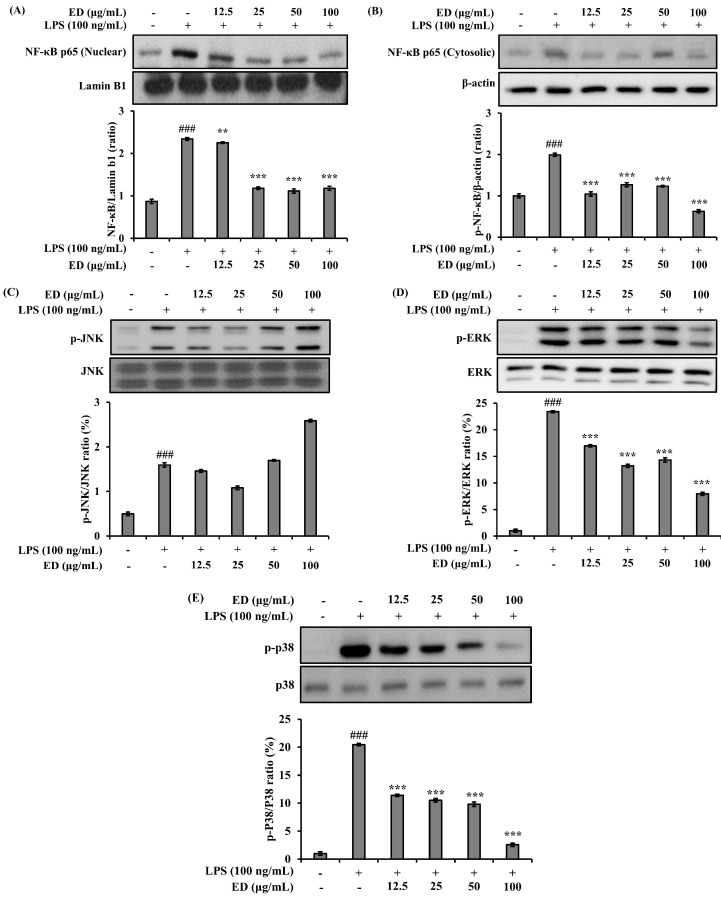
Effects of ED on the phosphorylation of (**A**,**B**) NF-κB and MAPK, (**C**) JNK, (**D**) ERK, and (**E**) p38 in RAW 264.7 cells. Lamin B1 and β-actin were detected and used as internal controls. The relative protein levels of p-JNK, p-ERK, and p-p38 were quantified using the Image J program and normalized to total JNK, ERK, and p38, respectively. The average value of three independent experiments is shown. All data are expressed as the mean ± SD of the experiment. ### *p* < 0.001 compared to the control group; ** *p* < 0.01 and *** *p* < 0.001 compared to the LPS control group. See also Appendix A.

**Figure 4 pharmaceuticals-16-01185-f004:**
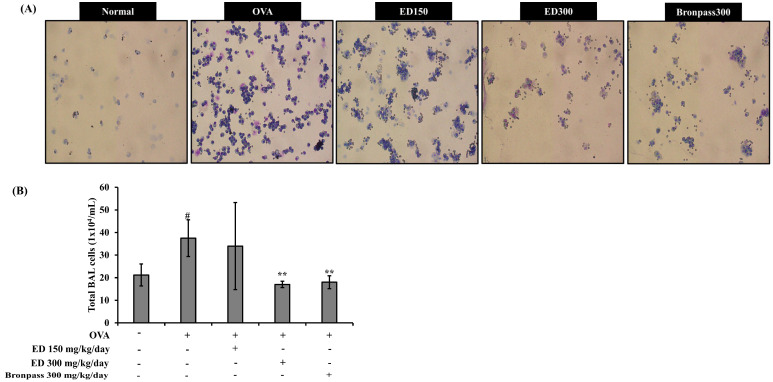
Effects of ED treatment on total cell count in OVA-induced asthmatic mice. (**A**) Diff-Quick staining (magnification ×200) of cells in BALF. (**B**) BALF was centrifuged and the pellet was assessed for total cell count using the Trypan blue dye exclusion test and differential count on cytospin slides using Diff-Quick staining. All results are shown as the mean ± SD (*n* = 5 per group). # *p* < 0.05 compared to the normal group. ** *p* < 0.01, compared to the OVA group.

**Figure 5 pharmaceuticals-16-01185-f005:**
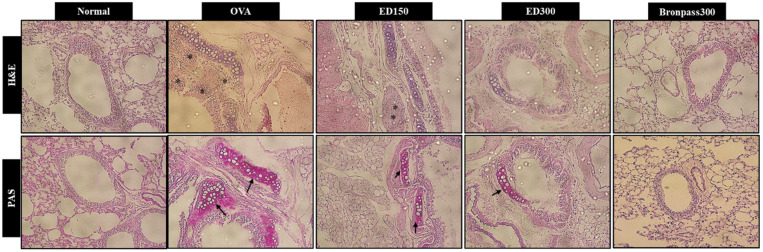
Effects of ED on histology of lung tissue in OVA-induced asthma mice. The lungs were stained using H&E (×200) and PAS (×200). *: Inflammatory cell infiltration, ↑: mucus secretion.

**Figure 6 pharmaceuticals-16-01185-f006:**
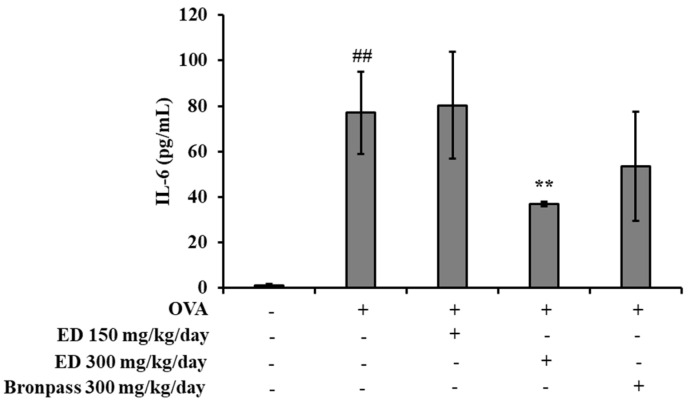
The effects of ED on serum level of IL-6 in OVA-induced asthmatic mice. All results are shown as the mean ± SD (*n* = 5 per group). *## p* < 0.01 compared to the normal group. *** p* < 0.01, compared to the OVA group.

**Figure 7 pharmaceuticals-16-01185-f007:**
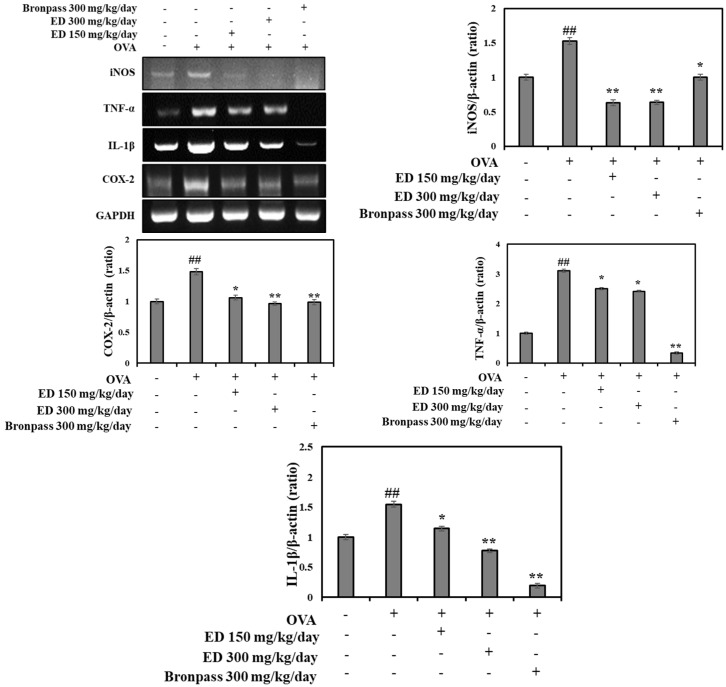
Effect of ED on the mRNA levels of cytokines in the lungs of OVA-induced asthmatic mice. RT-PCR analyses analysis using GAPDH as the loading control was performed for measurement of iNOS, TNF-α, IL-1β, and COX-2 mRNA expression in lung tissues. The relative mRNA levels of iNOS, TNF-α, IL-1β, and COX-2 were quantified using the Image J program. All results are shown as the mean ± SD (*n* = 5 per group). *## p* < 0.01 compared to the normal group. ** p* < 0.05 and *** p* < 0.01, compared to the OVA group.

**Figure 8 pharmaceuticals-16-01185-f008:**
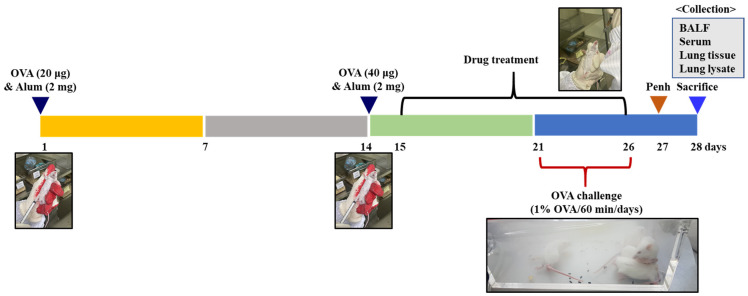
Experimental protocol. BALB/c mice were sensitized to OVA intraperitoneally on days 1 and 14 and challenged with aerosolized OVA on days 21–26. ED and Bronpass Tables were administered orally, starting from day 15, once daily for 12 days.

**Table 1 pharmaceuticals-16-01185-t001:** Conditions of HPLC analysis.

Content	Dieckol	Luteolin
Injection volume	5 μL	10 μL
Column temperature	25 °C	30 °C
Mobile phase	A: 0.01% acetic acid in DW (90%) + MeOH (10%)	A: 0.1% trifluoroacetic acid in DW
	B: Methanol	B: Methanol
Flow rate	0.7 mL/min	0.8 mL/min
Detection wavelength	230 nm	345 nm
Purity of the standards	95.6%	99%

**Table 2 pharmaceuticals-16-01185-t002:** Primer sequences.

Gene	Size (bp)	Sequence
iNOS	369	Sense 5′-CTTGCAAGTCCAAGTCTTGC-3′ Antisense 5′-GTATGTGTCTGCAGATGTGCTG-3′
TNF-α	390	Sense 5’-TTCGAGTGACAAGCCTGTAGC-3′ Antisense 5’-AGATTGACCTCAGCGCTGAGT-3′
IL-1β	385	Sense 5’-CATATGAGCTGAAAGCTCTCCA-3′ Antisense 5′-GACACAGATTCCATGGTGAAGTC-3′
COX-2	414	Sense 5′-ACATCCCTGAGAACCTGCAGT-3′ Antisense 5′-CCAGGAGGATGGAGTTGTTGT-3′
GAPDH	378	Sense 5’-CCAGTATGACTCCACTCACG-3′ Antisense 5’-CCTTCCACAATGCCAAGTT-3′

## Data Availability

Data is contained within the article and Appendix A.

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
