# Peer review of "ED Formula, a Complex of Ecklonia cava and Chrysanthemum indicum, Ameliorates Airway Inflammation in Lipopolysaccharide-Stimulated RAW Macrophages and Ovalbumin-Induced Asthma Mouse Model"

_pharmaceuticals, 2023, doi:10.3390/ph16081185_

Round 1
Reviewer 1 Report
The present paper reports the airway protective effects of ED, a formulation of Ecklonia cava and Chrysanthemum indicum, which have natural anti-inflammatory properties, both in vitro and in vivo. Through HPLC analysis, the major components of ED, dieckol and luteolin, were identified. As observed, ED exhibited no cytotoxicity and reduced NO production in LPS-induced RAW 264.7 cells. Additionally, ED downregulated COX-2 expression through the MAPK signaling pathway in LPS-induced RAW 264.7 cells. In an ovalbumin-induced asthma model, the ED-treated group showed decreased levels of inflammatory cytokines in lung tissue. Furthermore, the ED-treated group exhibited a reduction in the number of inflammatory cells in BALF and a decrease in serum IL-6 levels compared to the ovalbumin-treated group. In the opinion of this reviewer, this study was well conducted and reports interesting results. Therefore, I suggest its acceptance on Pharmaceutics after minor revision, as follow:
1. Figure 1 – please include the complete chromatogram obtained by HPLC analysis and include the structures of each compound.
2. Why not the minor compounds detected by HPLC were identified? As a formulation of different plants were evaluated, the complete characterization must be performed.
3. How was confirmed the identity of dieckol and luteolin? MS spectra were registered? Or standard compounds were used (co-elution in HPLC)?
4. Were the effects of ED compared with those obtained from purified dieckol and luteolin? Some discussion about this point could be included in the revised version of this manuscript.
Reviewer 2 Report
The manuscript “Ecklonia cava and Chrysanthemum indicum Linne Formula, ED, Ameliorates Airway Inflammation in LPS-Induced RAW Macrophages and OVA-Induced Asthma Mouse Model” investigated the effects of a formula composed of E. cava and C. indicum on airway inflammation in an ovalbumin (OVA)-induced asthma mouse model. However, this work is interesting and useful in the medical and pharmaceutical fields. I have the following suggestion to improve the quality:
1. Correct the title to: Ecklonia cava and Chrysanthemum indicum Formula, ED, Ameliorates Airway Inflammation in LPS-Induced RAW Macrophages and OVA-Induced Asthma Mouse Model.
2. The Abstract part must include clear general background, objectives, used methods, results and conclusion shortly so rephrase your abstract part to include these parts and avoid the use of abbreviations in the abstract as possible.
3. Improve your Keywords: Ecklonia cava; Chrysanthemum indicum Linne; Airway Inflammation; Asthma; Dieckol ; to be Ecklonia cava; Chrysanthemum indicum; Airway Inflammation; Asthma; Dieckol; Mouse models
4. Line 44 the word “Linne” must be not in italic.
5. There is a repetition in the introduction and discussion parts avoid that kindly.
6. The result part is well written and nothing to change.
7. Improve your discussion part by comparing your study outcomes with previously conducted studies with the same objectives.
8. In the material and method part add a section separate containing when you brought the used natural products, who identified them (the botanist's name), and where deposited the sample and voucher specimen codes.
9. The whole manuscript needs grammar, typos and editing corrections by a native speaker specialist in biomedical sciences.
10. If possible replace the used analytical method (HPLC) with a more accurate one as LC-MS-MS
The whole manuscript needs grammar, typos and editing corrections by a native speaker specialist in biomedical sciences.
Reviewer 3 Report
Article: pharmaceuticals-2532236 Title: Ecklonia cava and Chrysanthemum indicum Linne Formula, ED, Ameliorates Airway Inflammation in LPS-Induced RAW 264.7 Macrophages and OVA-Induced Asthma Mouse Model Recommendation: Majorchanges Comments The authors in the paper "Ecklonia cava and Chrysanthemum indicum Linne Formula, ED, Ameliorates Airway Inflammation in LPS-Induced RAW 264.7 Macrophages and OVA-Induced Asthma Mouse Model" investigated the airway protective effects of ED, a formulation of Ecklonia cava and Chrysanthemum indicum Linne, providing the data regarding their natural anti-inflammatory properties, both in vitro and in vivo. Authors should provide more information in Introduction section regarding ED, what were the starting points for authors to choose this brown alga and the plant. Are there some data that those two were used in traditional medicine for investigated effect? Please, explain in Materials and methods, page 9, lines 273-275 "E. cava ethanol extract 273 (Recognition No. 2015-6) was purified from the ethanol extract of the brown alga E. cava". What is the ratio of used extracts in ED formulation? Please, give the full HPLC chromatograms of E. cava and C. indicum, as well of their mixture (used ED formulation in the experiments) Please, give the purity of the standards used in HPLC analysis, page 9, lines 281, 282. Please, explain the procedure of HPLC analysis of ED formulation, as in the paper page 10, Table 1 there were only data regarding the used standards... Please, give in the Material and Method section the used concentrations of the investigated samples in the experiment (page 10, line 300: the cultured cells were treated with ED, Dieckol, or Luteolin for 24 h; line 312) Line 352, please explain what the criteria were to choose the concentration of ED 150 and 300 mg/kg in experiments. How were the administrated samples prepared? What was the solvent? Please, explain why only the experiments which results were presented in Figure 2. were performed for all ED, dieckol, and luteolin, and in the rest of the experiments the correlation between the observed activities of used ED and the determined actives in the mixture (luteolin and dieckol) was not established? Please, explain the chosen concentration of luteolin and dieckol in the experiments presented in Figure 2.Author Response
Please see the attachment.

Reviewer 4 Report
1) The animals – The number of the agreement of the Bioethics Commission should be provided concerning animal studies.
2) Purity of the used standards should be given.
3) The validation data concerning HPLC quantitative analysis should be presented; LOD,LOQ, calibration curves for standards, recovery studies. Statistics for HPLC quantitative analysis should be given (not only for biological studies).
4) ED is standardized for dieckol (60 mg/g) – was it standardized for luteolin also? Or luteolin containing extract was connected/mixed with ED ? It is not clearly described.
5) As the effects of ED, and also of dieckol and luteolin as the single compounds were tested it is interesting the quantitative amount of dieckol and luteolin in ED in the tested concentrations of ED (25 µg of ED on mL, 50, 100, 200, 400) see lines 74-76.
As we have (see Figure 1) 9.5 mg/g ED of dieckol and 1.5 mg/g ED luteolin, so we expect to have; in 100 ug ED; 0.95 µg of dieckol and 0.15 µg of luteolin ; for 25 ug ED it will be ; c.a. 0.24 ug of dieckol and 0.04 ug of luteolin. Am I right ? What constituted the whole amount of ED formula, as dieckol constituted 0.95% only and luteolin 0.15 % only of the ED ? Are dieckol and luteolin in fact the main constituents of the ED ?
6) Line 70 should be; (unit mg/g ED)
7) Line 77-78; should be; “According to the results, ED and its main active compounds tested separately had no cytotoxic effects I any of the concentrations tested.”
8) Line 54; should be “…as the ED compounds” ?
9) Line 61; is “utilized” should be “considered”
10) Abstract line 11; in vivo should be in italics
11) Abstract line 18 should be; “treatment of…” not “improving ”
Round 2
Reviewer 2 Report
The authors conducted all the required corrections